# Olfactory bulb differently synchronizes ventral hippocampus–medial prefrontal cortex circuit during spatial working memory across social dominance hierarchies

Elham Bakhshi Jifroudi[1], Soomaayeh Heysieattalab[1]*, Farhad Farkhondeh Tale Navi[1], Ali Jaafari suha[2], Faezeh Zarfsaz[1], Yousef Panahi[3]

1 Department of Cognitive Neuroscience, Faculty of Education and Psychology, University of Tabriz, Tabriz, Iran, 2 Department of Physiology, School of Medicine, Shahid Beheshti University of Medical Sciences, Tehran, Iran, 3 Department of Basic Sciences, Faculty of Veterinary Medicine, University of Tabriz, Tabriz, Iran

* heysieattalab@gmail.com, heysieattalab@tabrizu.ac.ir

## Abstract

The olfactory system plays a central role in social communications in most mammals, especially social dominance hierarchies (SDH). The olfactory bulb (OB) is intricately linked to cortical and limbic areas concerned with social and learning and memory functions. To elucidate the neural underpinnings of cognitive performance in dominance hierarchies, this study investigated how OB delta oscillations (< 4 Hz) synchronize gamma oscillatory activity (30–50 Hz) within the ventral hippocampus-medial prefrontal cortex (vHPC-mPFC) circuit in rats of varying social ranks. Twenty-one male Wistar rats were home-caged in natal triads and categorized as dominant, middle-ranked, or subordinate based on tube test performance. After a month of social cohabitation, rats underwent a spatial working memory (SWM) task in T-maze with two delay intervals: an easy level of task (30-second delay) and a difficult level of task (5-minute delay). The percentage of correct responses showed no significant difference between social ranks. However, subordinates showed lower latency in reaching the goal arm, while middle-ranked rats exhibited longer latency in the 30-second delay. Electrophysiological results revealed higher delta power spectral densities (PSDs) in OB during correct responses of the easy level of task in dominant and subordinate groups. Also, subordinates showed overall higher gamma coherence in the mPFC-vHPC circuit in positive correlation with delta PSD of OB that can be related to decreased latency during correct responses in the easy level of task. These findings highlight the engagement of the olfactory system in cognitive processes regarding social rank.

**Data availability statement:** All relevant data for this study are publicly available from the OSF repository (https://osf.io/35qck).

**Funding:** The author(s) received no specific funding for this work.

**Competing interests:** The authors have declared that no competing interests exist.

## 1. Introduction

Social cognition encompasses the processing of social cues that guide learning and memory in social environment. These cues include facial expressions, like fear and disgust, that signal emotional states, gaze direction toward the salient stimuli [1,2], and the rank in social dominance hierarchies (SDH) that may influence the perception of these cues [3]. Dominance hierarchy is a widespread structure that influence survival, physical and mental well-being, overall quality of life, as well as memory and social decision-making [4–8]. Neural underpinnings of SDH and their impact on effective performance in demanding situations are vital for successful behavioral strategies [9,10].

Exploring brain regions involved in social interactions is crucial for cognitive and emotional functions. The olfactory bulb (OB) plays a key role in social behaviors [11], and olfactory input is necessary for establishing dominance relationships [12–14]. The OB is connected to key cortical and subcortical regions involved in SDH and cognitive functions. Its oscillations modulate rhythmic activity in neocortical and limbic areas [15]. The OB projects to the entorhinal cortex (EC), which communicates bidirectionally with the hippocampus (HPC) [16], and also interacts with the medial prefrontal cortex (mPFC) via the anterior olfactory nucleus (AON) [17,18]. Olfactory oscillations can influence mPFC activity during social behaviors [16]. The mPFC is extensively connected with subcortical areas like the HPC, and their synchronous activity is crucial for memory processes, particularly working memory. The mPFC receives event-related information from the ventral HPC (vHPC), including social status and spatial details [19]. Spatial working memory (SWM) allows temporary storage of spatial information to guide decisions and achieve goals [20,21].

Several studies report conflicting results on social status and SWM. While some research suggests that dominant animals exhibit better SWM, other findings indicate the opposite [21,22]. For example, in home-caged sibling rats, subordinate rats performed better than dominant rats in the Morris water maze (MWM), but no difference appeared in the Y-maze [23]. These discrepancies may stem from differences in cognitive load, which depends on the delay between cue detection and response [24]. In this context, our previous study examining the vHPC–mPFC circuit in relation to SWM showed that differences between easy and difficult degrees of task at both behavioral and electrophysiological levels, highlighting the importance of task demands in working memory [25]. In addition to the PFC and HPC areas, delta oscillations (< 4 Hz) in the OB have been shown to be essential for proper working memory function [26–29]. For example, hippocampal respiratory rhythm (HRR) and prefrontal respiratory rhythm (PRR), both driven by the olfactory system, support cognitive integration in the HPC and PFC [29–32]. OB deletion impairs working memory [33], and its delta activity coordinates vHPC–mPFC coherence in the gamma range, critical for SWM performance [15]. Recently, it has been proposed that the interaction between theta (4–10 Hz) and low gamma (30–50 Hz) oscillations in the hippocampus is involved in memory retrieval. [34,35]. This band is thought to be key in coordinating the activity between spatially distributed neuronal ensembles, which is crucial for complex cognitive functions such as those involved in our T-maze task [35]. We

focused on delta and low gamma ranges due to their increased power spectral density (PSD) and coherence during SWM [15,36].

Although the OB plays a central role in the formation of SDH is well established in the existing literature, its precise interaction with the VHPC-mPFC circuit during SWM processes, particularly across different social ranks, remains largely unknown. Understanding this interaction is important given that cognitive load, or the amount of mental effort required during SWM task, can vary significantly depending on the social context and investigating how these brain regions coordinate under different cognitive loads provides valuable insight into the neural mechanisms underlying SDH and working memory. This represents a significant gap in our understanding of how social cues, processed by the OB, influence higher order cognitive functions related to behavior and SDH. Previous research has often focused on the role of the OB in direct olfactory processing or its impact on social cognition. Our study uniquely extends this issue by examining its synchronization role in a key circuit for social cognition and working memory, and, importantly, how this synchronization varies based on different social status (dominant, middle, or subordinate rank) and cognitive loads (T-maze with two delay intervals: an easy level (30-second delay) and a difficult level of task (5-minute delay)). Moreover, the study also reports the power and cross-area coherence of functional oscillations in these key brain regions for SWM and SDH, namely OB, VHPC, and mPFC.

## 2. Materials and methods

### 2.1. Animals and ethical statement

A total of 21 rats were included in this study, consisting of 3 male littermates from each of 7 mothers. After weaning, they were housed in their home-cages as triads of littermates from the same mother, matched for weight (70–80 g), and kept under identical conditions (7 cages) at 21±2 °C with a 12-hour light–dark cycle (from 8 am to 8 pm). The 4-week-old rats, weighing 70–80 g, were obtained from Urmia Medical Sciences University and maintained in standard animal research facilities with ad libitum access to food and water. The investigation protocol was consented to by the "Research Ethics Committee of Tabriz University (IR.TABRIZU.REC.1401.050)", in compliance with established guidelines for the ethical treatment of animals in research.

### 2.2. Study design

The study was carried out in two consecutive stages: a behavioral phase and an electrophysiological phase. During the ninth postnatal week, each triad was familiarized with the tube-test procedure, and from weeks 10–12 their relative dominance hierarchy was established based on repeated tube-test encounters. The period when male rats transition from adolescence to adulthood, marked by significant aggressive interactions among cage-mates, was identified as the time for the development of SHD [37]. After maintaining this hierarchy for approximately one month, animals were subjected to surgery at 18–24 weeks of age. Following a seven-day recovery, they underwent the SWM task in a T-maze while neural recordings were obtained. The protocol was conducted in a reward-free manner because dominant and subordinate rats may differ in their food motivation, ensuring that observed performance in the SWM task reflected cognitive ability rather than differences in appetite [38]. Task difficulty was manipulated using two delay intervals between the sample and choice runs: the easy condition (30 s delay) and the difficult condition (5 min delay). Increasing the delay length raised the cognitive demand, allowing potential differences between dominant and subordinate rats to be more clearly observed. At this stage, neuronal activity within the OB–vHPC–mPFC circuits was monitored during task execution, providing electrophysiological insight into how SDH development influences memory-related network dynamics (Fig 1).

**2.2.1. Behavioral phase. 2.2.1.1. Social dominance tube test**. The tube test, a well-established method for assessing SDH [39], employs a translucent Plexiglas tube (1 m in length and 5.4 cm in internal diameter), designed to permit the passage of a single adult rat without allowing it to turn around inside the tube [23]. The test consists of two stages: training (3 days), and testing (5 days). In the training phase, the rats are allowed to explore the apparatus for one

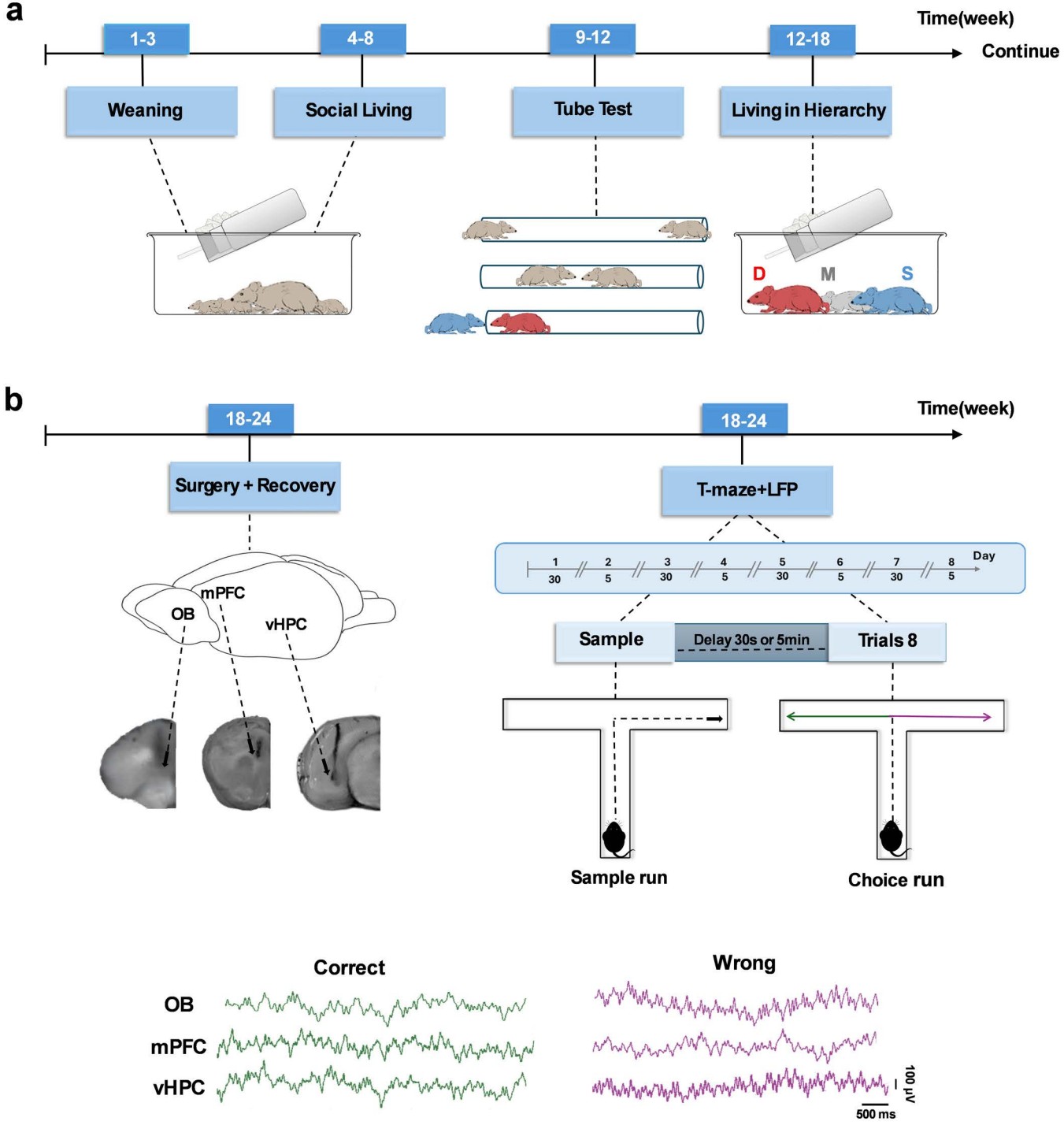

**Fig 1. Overview of the experimental timeline for behavioral and electrophysiological procedures. (a)** Social development stages of the three littermates before and after the tube test: Dominant (red), Middle-ranked (gray), and Subordinate (blue). **(b)** Surgical timeline and electrophysiological recording. Electrodes were implanted in the OB, mPFC, and vHPC. After seven days of recovery, rats performed the SWM task, which included two levels of difficulty (easy: 30 s; difficult: 5 min), while LFP signals were recorded. Representative histological confirmation of each brain region and sample LFP traces for correct and wrong trials are shown. (LFP: local field potential; OB: olfactory bulb; mPFC: medial prefrontal cortex; vHPC: ventral hippocampus; SWM: spatial working memory).

minute. Each training session consisted of eight movements forward movements to the other side of the tube, balanced for the right or left starting side. On test trial days, each pair of rats was simultaneously released into the tube from opposite ends. When they met in the middle, the subordinate rat was pushed out by the dominant rat. The rat that forced the other to retreat completely out of the tube received one point, while the retreating rat received zero. This procedure was repeated with the entry ends of the rats alternated in each trial. Each cage underwent three test trials per day, with every rat competing against the other two (e.g., A-B, A-C, B-C). The average number of wins across five testing days was used to determine the relative SDH within each cage, ranked as dominant (D), middle-ranked (M), and subordinate (S) [39,40]; (Fig 1a). If no winner emerged within two minutes (i.e., a tie) or both rats retreated simultaneously, the trial was repeated. Additionally, when changing the entry side produced different outcomes, the trial was repeated until one rat was declared the winner or each rat received 0.5 points. The tube was cleaned with 70% ethanol after each trial.

**2.2.1.2. Spontaneous alternation in T-maze test**. The T-maze is a commonly used apparatus in cognitive animal research, providing a straightforward choice between the left and right arms. The maze is a raised, T-shaped structure oriented horizontally, consisting of a central arm 55 cm in length and two identical target arms, each 50 cm long, 10 cm wide, and with 15-cm-high walls. Each arm is equipped with a transparent guillotine door. The T-maze allows the assessment of memory function and spatial learning under various stimuli [41]. Unlike protocols using food rewards, the spontaneous alternation version of the T-maze does not require habituation and is conducted without reinforcement. This protocol was selected to account for potential differences in food motivation between dominant and subordinate rats [38]. In this task, it is the novelty of the maze arms that drives spontaneous exploration [38]. During the sample phase, each rat was initially placed at the starting point, located at the base of the central arm perpendicular to the left and right arms, and allowed to choose one of the two arms. After entering a target arm, the rat was confined there for 30 seconds before being returned to the starting point. A delay interval between the sample and choice phases was implemented according to task difficulty: 30 s for the easy condition and 5 min for the difficult condition. Rats could freely select the target arm in both phases [42]. Task difficulty was alternated daily over eight days (four days at 30 s delay, four days at 5 min delay), with six trials per day (one sample trial and five choice trials). Each trial had a 180-second time limit for maze exploration [22]. To minimize olfactory cues, the arms were cleaned with 70% ethanol between the sample and choice phases. Animal movements were recorded via an overhead camera for subsequent behavioral analysis. All experiments were performed daily in the afternoon between 3:00 PM and 6:00 PM, with light intensity maintained at 100 lux (Fig 1b).

**2.2.2. Electrophysiological phase. 2.2.2.1. Electrode implantation and histological verification**. To implant the electrodes in the specified areas, rats were first anesthetized with a combination of ketamine (100 mg/kg) and Xylazine (10 mg/kg). The anesthetized animals were then placed in a stereotaxic frame (Toosbioresearch, Mashhad, Iran), and a heating pad was used to maintain body temperature at 37 °C throughout the surgery. Anesthesia depth was continuously monitored by assessing tail and paw pinch reflexes. Additionally, vitamin A ointment was applied to the eyes to prevent corneal drying during surgery. Local anesthesia of the scalp was provided by subcutaneous injection of 0.5 ml Persocaine to minimize pain during the incision. Following drilling the skull, stainless-steel electrodes (127 µm total diameter of the *coated* electrode, A.M. System Inc., USA) were unilaterally implanted into stereotaxic coordinates of OB (AP: +8.5 mm; ML: −1 mm; DV: −1.5 mm), mPFC (AP: +3.2 mm; ML: −0.6 mm; DV: −3.6 mm) and vHPC (AP: −4.92 mm; ML: −5.5 mm; DV: −7.5 mm) according to the rat brain atlas (Paxinos and Watson, 2007). One additional hole was drilled for the reference electrode and dental adhesive was applied on the skull to fix the electrodes and head-stage. After the completion of experiments, to ensure accurate electrode placement, rats were deeply anesthetized with carbon dioxide. Once breathing ceased completely and the animals exhibited ocular pallor, they were rapidly decapitated using a guillotine. Brains were carefully extracted and fixed in 4% paraformaldehyde at 4 °C for 48 h. Subsequently, the brains were sectioned using a vibroslicer, stained with methylene blue, and examined under a microscope (AC 230V 50 Hz, Fig 1b).

**2.2.2.2. Electrophysiological recordings**. One-week post-surgery, local field potential (LFP) was recorded from the OB, mPFC, and CA1 area in vHPC. Each rat underwent LFP recording (BIODAC-A, TRITA Health Tec., Tehran, Iran)

during the SWM task for eight consecutive days. Since the spontaneous selection in the SWM task, the rats did not require prior training or habituation to the maze space. 30 min before recording commencement, the rats were transferred from the animal house to the laboratory to get used to the space inside the laboratory and the recording chamber which provided isolation from sound and environmental noises. Subsequently, they were moved into the room and were positioned at the starting point of the maze. Recordings were conducted using a digital head-stage probe capable of simultaneously recording from three brain regions. Frequency settings were adjusted to capture waves below 250 Hz. Throughout the task execution, all steps were recorded via a camera located on the top of the chamber. In Fig 1b, see images of the testing steps and comparison of raw data of correct and wrong responses.

## 2.3. Extracting data

**2.3.1. Extracting behavioural data.** The percentage of correct responses and the time it takes for a rat to reach the goal arm (latency) in the T-maze from the starting point of the T-maze during the SWM task were determined by careful analysis of video recordings of behavioural sessions. [43].

**2.3.2. Extracting electrophysiological data. 2.3.2.1. Signal processing.** A tailored program created with MATLAB version 2022b (MathWorks, Natick, MA, USA) was used to analyze LFPs recorded during feedback sessions. To analyze the signals, for each specific band, the relevant filter was selected. We filtered the LFP recorded from OB in the delta range by a bandpass filter (0.5–4 Hz). We then assessed the mean power spectral densities (PSDs) of the LFP for each trial. The Hanning window was multiplied by the raw LFP for a given trial, and the PSD was computed with the *"p-Welch"* function. The frequency resolution of the resultant PSD was 0.05 Hz. A high-pass filter with a 0.5 Hz cutoff was applied to the raw LFP to remove low-frequency baseline drift caused by movement, a standard procedure for artifact reduction in rodent electrophysiology [44].

To determine the coherence between OB-mPFC, OB-vHPC, and vHPC-mPFC, we calculated magnitude-squared coherence using MATLAB's *"mscohere"* function. Mean coherency values in the gamma range (30–50 Hz) were obtained for each time interval. The results represent the average number of correct and wrong responses at two difficulty levels (30s and 5 min), calculated from four alternating sessions each. Thus, the electrophysiological data presented represents the analysis of LFP over 8 consecutive days in different groups.

The key point to be examined and analyzed was the decision point, which refers to the moment when one of the left or right arms is chosen, plus 3 seconds before and 1 second after this moment; overall 4 seconds.

## 2.4. Statistical analysis

Two-way repeated measures ANOVA was used to compare the differences between groups. Where relevant, Bonferroni tests and post hoc analyses were also performed. Additionally, multiple linear regression and linear regression analyses were employed to explore correlations between variables across different groups. All data are presented as mean±SEM, with *p-values* below 0.05 were considered statistically significant. We first analyzed the groups based on correct and wrong responses at two levels of difficulty in the T-maze task. Subsequently, we examined the correct responses within the groups across the two difficulty levels of the T-maze task. Two levels of cognitive load were used in the T-maze task to more accurately examine the effects of SDH on SWM. Using these two levels allows for comparison of performance under both easy and difficult levels of task, providing greater sensitivity in detecting differences related to social factors. Statistical analysis was carried out using GraphPad Prism Version 9.5.1 (GraphPad Software, Boston, Massachusetts USA, www.graphpad.com).

## 3. Results

### 3.1. Behavioral results

**3.1.1. Social dominance tube test.** The relative dominance of each rat was established based on outcomes of the social dominance tube test. With seven cages containing three rats each, we calculated the frequency of wins per session for each rank. Dominant rats typically achieved 2 wins per session, middle-ranked rats generally secured 1 win

per session, while subordinate rats frequently had 0 wins in each session. The final ordinal dominance status for each cage was determined by averaging the total number of wins across all experimental days, ranked in descending order as dominant (D; rank 1), middle-ranked (M; rank 2), and subordinate (S; rank 3), respectively (D: $1.84 \pm 0.06$, M: $0.85 \pm 0.06$, S: $0.25 \pm 0.07$ for all cages; Fig 2a). A one-way ANOVA was conducted to compare the number of wins, revealing a significant difference among the three social ranks ($F_{(2, 102)} = 139.20$, $P < 0.0001$). Bonferroni's multiple comparisons tests showed significant differences between dominant and subordinate ($t = 16.47$, $P < 0.0001$); dominant and middle-ranked ($t = 10.54$, $P < 0.0001$) and subordinate and middle-ranked group ($t = 5.94$, $P < 0.0001$; Fig 2b).

**3.1.2. Percentage of correct responses in the easy and difficult levels of SWM task across different social ranks.** The percentage of correct responses in the choice run was determined for each session across three social ranks based on the T-maze task performance. We used the two-way ANOVA to compare the percentage of correct responses between the *two levels of task difficulty* for each rank. There was no significant main effect of the group ($F_{(2, 36)} = 1.14$, $P = 0.32$), task difficulty ($F_{(1, 36)} = 1.52$, $P = 0.22$), and interaction effect ($F_{(2, 36)} = 0.47$, $P = 0.62$; Fig 3a).

**3.1.3. Latency of correct responses in the easy and difficult levels of SWM task across different social ranks.** Comparison of latency between the *two levels of task difficulty* for each rank using two-way ANOVA revealed significant main effect of group ($F_{(2, 36)} = 8.69$, $P < 0.01$), and task difficulty ($F_{(1, 36)} = 5.13$, $P < 0.05$). There was no significant interaction effect ($F_{(2, 36)} = 1.96$, $P = 0.15$; Fig 3b). Bonferroni's multiple comparisons test revealed a significant difference only between subordinates and middle ($t_{(36)} = 4.28$, $P < 0.01$; Fig 3b). Furthermore, on the easy level of task, the middle-ranked group displayed higher latency ($M = 80.24 \pm 9.54$) compared to dominants ($M = 69.16 \pm 8.16$), with a large effect size (Cohen's $d = 1.25$), though the difference was not significant after Bonferroni correction ($P = 0.10$). These findings show that the subordinates had the shortest latency, while the middle-ranked rats had the longest latency during all sessions. Also, social ranks showed similar delay times when choosing the target arm in the difficult level of task.

### 3.2. Electrophysiology results

Although behavioral accuracy did not differ by social rank, we analyzed brain activity to see if similar performance stemmed from different neural strategies under varying cognitive demands.

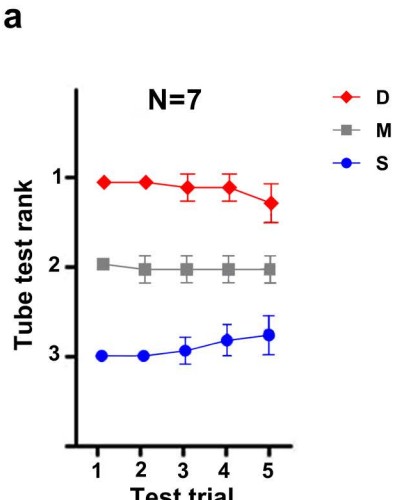

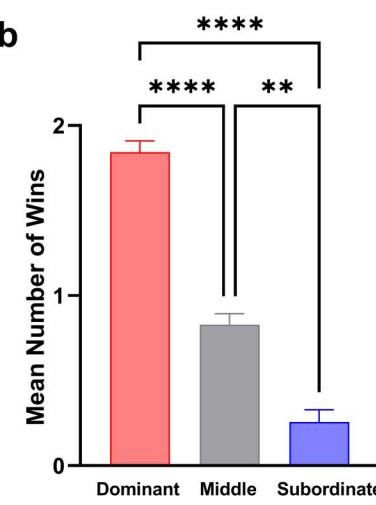

**Fig 2. Determining relative dominance using the tube test. (a)** Summary graph for 7 cages measured. The average rank belonging to each rank category from the first test trial was calculated. D: Dominant, M: Middle-ranked, S: Subordinate, N: Number of cages **(b)** The average number of wins in the Tube test for the three groups. **p<0.01, ****p<0.0001; Values express as the mean ±SEM.

**a**
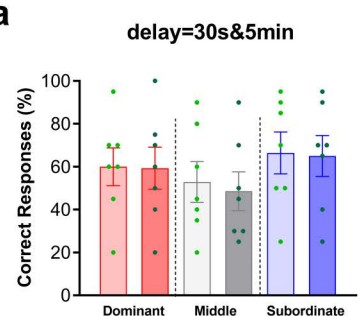

**b**
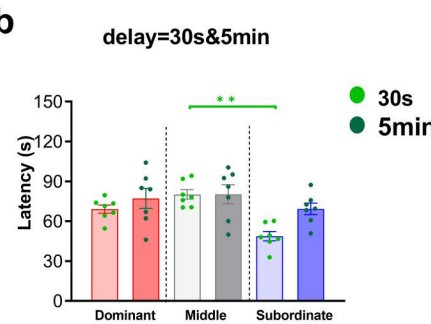

**Fig 3. T-maze spontaneous alternation test in two task difficulty. (a)** Comparison of percentage of correct responses between two task difficulty within each group for short (30 s, light colors) and long (5 min, dark colors) delays. **(b)** Group comparison of latency in two levels of task difficulty (delay = 30s&5min); **p < 0.01, Values are expressed as mean ±SEM.

**3.2.1. OB delta PSD during correct and wrong responses in the easy and difficult levels of SWM task across different social ranks.** We investigated changes in OB delta PSD (<4 Hz) across different social ranks during correct and wrong responses in *two difficulty levels* of the T-maze task. The samples of raw LFP signals and time-frequency spectrograms from the OB during correct and wrong trials at *both difficulty levels* are shown in Fig 4a–d. In the *easy level of task,* two-way ANOVA revealed a significant difference between correct and wrong responses across social rank (F (1, 162) = 26.31, P<0.0001; Fig 4e). Bonferroni's multiple comparisons test showed significantly higher delta PSD of OB during correct responses in dominants (t (162) = 3.29, P<0.05) and subordinates (t (162) = 4.39, P<0.001; Fig 4e), suggesting enhanced OB delta activity when these groups made correct choices However, further comparison between social ranks during *correct or wrong responses* showed no significant main effect of group (F (2, 162) = 1.84, P=0.16) and no significant interaction effect (F (2, 162) = 2.64, P=0.07; Fig 4e).

In the *difficult level of task*, two-way ANOVA revealed no significant main effects of group (F (2, 162) = 0.03, P=0.96) or response type (F (1, 162) = 0.23, P=0.63), and no significant interaction effect (F (2, 162) = 1.03, P=0.35; Fig 4f), indicating similar OB delta PSD patterns across ranks regardless of response correctness.

When comparing the OB delta PSD between *easy* and *difficult levels of task* for each rank during correct responses, there were no significant main effects of group (F (2, 162) = 2.03, P=0.13) or interaction effects (F (2, 162) = 1.31, P=0.27). However, the main effect of task difficulty was significant (F (1, 162) = 4.62, P<0.05; Fig 4g), indicating overall higher OB delta PSD in the easy compared to the difficult level of task.

**3.2.2. Correlation between OB delta PSD and correct responses in the easy and difficult levels of SWM task across different social ranks.** A multiple regression analysis was conducted to examine the effect of correct response and social rank on the OB delta PSD across two *levels of task difficulty*. In *the easy level of task*, the overall regression model was significant (F (5, 78) = 3.37, P=0.008), as was the interaction effect between group and response type (F (2, 78) = 3.59, P=0.03), indicating that the combination of social rank and response type significantly predicted the OB delta PSD. However, neither the main effect of group (F (2, 78) = 2.07, P=0.13) nor response type alone (F (1, 78) = 0.70, P=0.40) reached statistical significance (Fig 4h). Correlation analysis revealed distinct patterns among ranks. In *the easy level of task*, OB delta PSD was positively correlated with the percentage of correct responses in subordinates (r=0.43, P=0.02), negatively correlated in middle-ranked rats (r=−0.48, P=0.009), and showed a non-significant negative trend in dominants (r=−0.16, P=0.40; Fig 4h).

In *the difficult level of task*, the overall regression model (F (5, 78) = 3.71, P=0.004), the effect of group (F (2, 78) = 6.32, P=0.002), the effect of response type (F (1, 78) = 10.11, P=0.002) and interaction effect between group and response type was also significant (F (2, 78) = 8.78, P=0.0004), indicated that both factors independently and

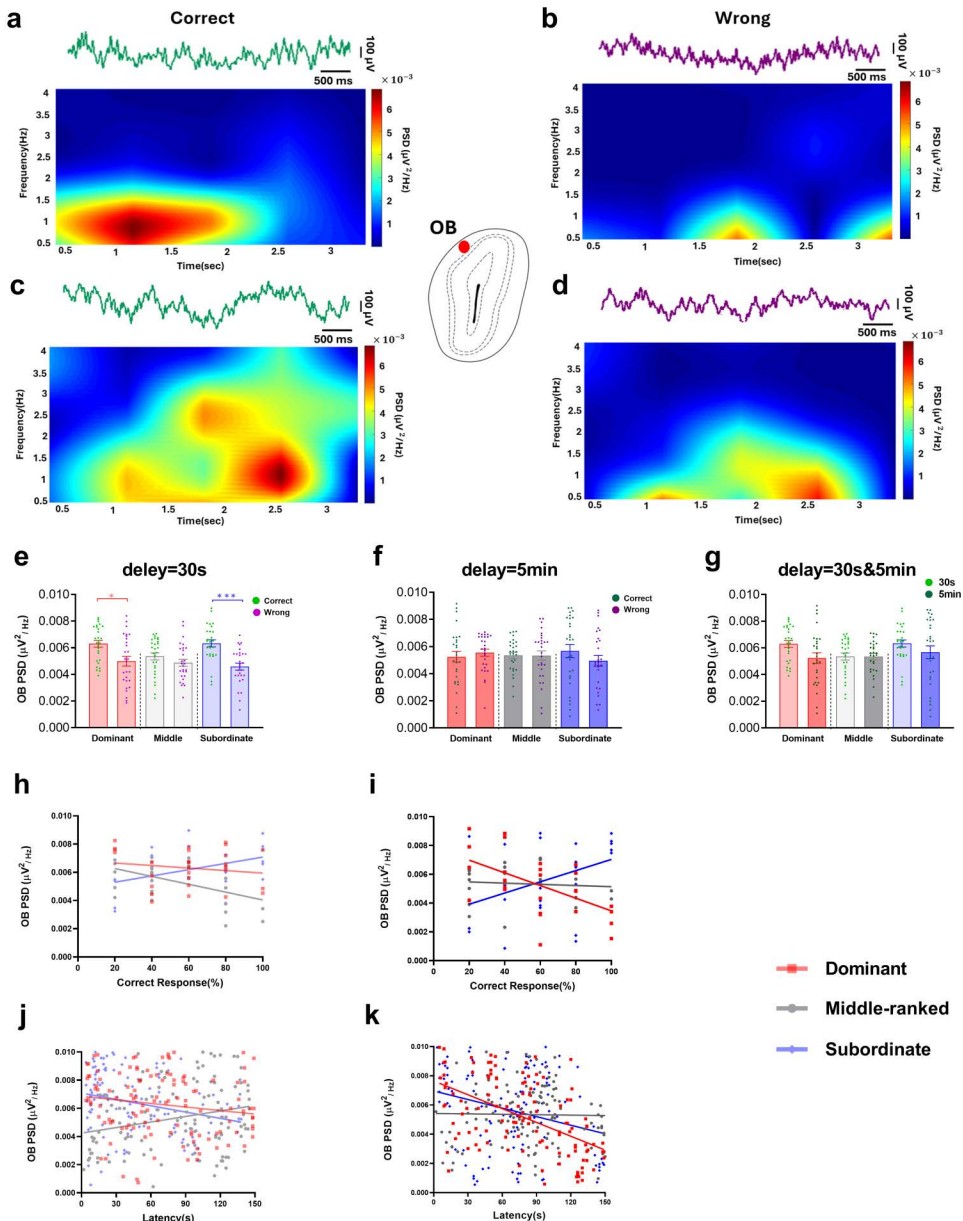

**Fig 4. Samples of time–frequency and raw LFP signals in OB during correct and wrong trials of the SWM task in the subordinate group (a–d):** **(a) correct trial, easy level of task; (b) wrong trial, easy level of task; (c) correct trial, difficult level of task; (d) wrong trial, difficult level of task.** OB delta PSD and responses in SWM task across social ranks (e–k): The average of the OB delta PSD of correct and wrong responses in dominant, middle-ranked, and subordinate groups during the (e) easy and (f) difficult levels of task, *$p < 0.05$, **$p < 0.001$; Values are expressed as mean ±SEM. (g) The comparative delta PSD of the OB area in the correct responses of two levels of task difficulty of the three groups. Respectively: dominant (red), middle-ranked (gray), and subordinate (blue). Correlation of OB delta PSD with the percentage of correct responses of three groups in 4 days during the (h) easy and (i) difficult levels of task. Correlation of OB delta PSD with latency of correct responses of three groups in 4 days in the (j) easy and (k) difficult levels of task. (OB: olfactory bulb; PSD: power spectral density).

interactively influenced OB delta PSD (Fig 4i). Correlation analysis showed that higher OB delta PSD was associated with more correct responses in subordinates (r = 0.38, P = 0.04), but fewer correct responses in dominants (r = −0.55, P = 0.002). No significant relationship was found for middle-ranked rats (r = −0.08, P = 0.66; Fig 4i). These findings suggest that in both task difficulties, OB delta PSD in subordinates positively correlated with correct responses, whereas in dominants the relationship was consistently negative.

**3.2.3. Correlation between OB delta PSD and latency during correct responses in the easy and difficult levels of SWM task across different social ranks.** A multiple regression analysis was conducted to examine the effect of latency and social rank on the OB delta PSD across two levels of task difficulty. In the easy level of task, the overall regression model was significant (F (5, 354) = 7.53, P < 0.0001), as were the main effect of group (F (2, 354) = 16.26, P < 0.0001) and the interaction effect between group and response type (F (2, 354) = 9.84, P < 0.0001), indicating that the combination of social rank and response type significantly predicted OB delta PSD. However, the main effect of response type alone did not reach significance (F (1, 354) = 3.31, P = 0.06; Fig 4j). Correlation analysis revealed a significant negative relationship between latency and OB delta PSD in subordinates (r = −0.27, P = 0.002) and a significant positive relationship in middle-ranked rats (r = 0.25, P = 0.005), while dominants showed a non-significant negative trend (r = −0.16, P = 0.07; Fig 4j). These results suggest that changes in the OB delta PSD with different latency are rank-dependent in the easy level of task.

In the difficult level of task, the overall regression model (F (5, 354) = 10.71, P < 0.0001) and the effect of social rank (F (2, 354) = 5.51, P = 0.004) were statistically significant. In addition, the effect of latency (F (1, 354) = 39.37, P < 0.0001) and interaction effect between group and latency was significant (F (2, 354) = 7.96, P = 0.0004; Fig 4k). As a result, the combination of group and latency significantly predicted delta PSD of OB. Correlation analysis showed a significant negative relationship between response type and OB delta PSD in both dominants (r = −0.52, P < 0.0001) and subordinates (r = −0.26, P = 0.003), with no significant relationship in middle-ranked rats (r = −0.01, P = 0.84; Fig 4k). Overall, these findings indicate that the influence of response type on the OB delta PSD is predicted by social rank, with consistent negative associations in dominants and subordinates across both difficulty levels.

**3.2.4. Gamma coherence in OB-mPFC, OB-vHPC, and vHPC-mPFC circuits during correct and wrong responses in the easy level of SWM task across different social ranks.** In *the easy level of task*, we analyzed gamma coherence in the OB-mPFC, OB-vHPC, and vHPC-mPFC circuits during correct and wrong responses across social ranks using two-way ANOVA.

**3.2.4.1. OB-mPFC coherence.** There were significant main effects of group (F (2,162) = 5.06, P < 0.01), response type (F (1,162) = 38.91, P < 0.0001), and their interaction (F (2,162) = 3.10, P < 0.05; Fig 5b). Post-hoc Bonferroni tests showed higher gamma coherence during correct vs. wrong responses in dominant (t (162) = 3.92, P < 0.01) and subordinate (t (162) = 5.17, P < 0.0001) rats. Subordinates also exhibited significantly higher coherence than middle-ranked rats during correct responses (t (162) = 3.94, P < 0.01; Fig 5b).

**3.2.4.2. OB-vHPC coherence.** Significant effects of group (F (2,162) = 7.78, P < 0.001), response type (F (1,162) = 192.7, P < 0.0001), and interaction (F (2,162) = 5.00, P < 0.01) were observed (Fig 5c). All ranks showed higher coherence in correct compared to wrong responses (dominant: t = 7.87; middle-ranked: t = 5.85; subordinate: t = 10.32; all P < 0.0001). Subordinates had higher coherence than middle-ranked rats during correct trials (t = 4.99, P < 0.0001), with no differences between dominants and other groups (Fig 5c).

**3.2.4.3. vHPC-mPFC coherence.** The main effects of group (F (2,162) = 51.03, P < 0.0001), response type (F (1,162) = 321.8, P < 0.0001), and interaction (F (2,162) = 29.03, P < 0.0001) were significant (Fig 5d). All ranks exhibited higher coherence during correct vs. wrong responses (dominant: t = 6.95; middle: t = 7.54; subordinate: t = 16.57; all P < 0.0001). Subordinates showed higher coherence than dominants (t = 6.77, P < 0.0001) and middle-ranked (t = 11.44, P < 0.0001) rats during correct trials. Dominants also had higher coherence than middle-ranked rats in both correct (t = 4.67, P < 0.0001) and wrong (t = 5.25, P < 0.0001; Fig 5d) responses.

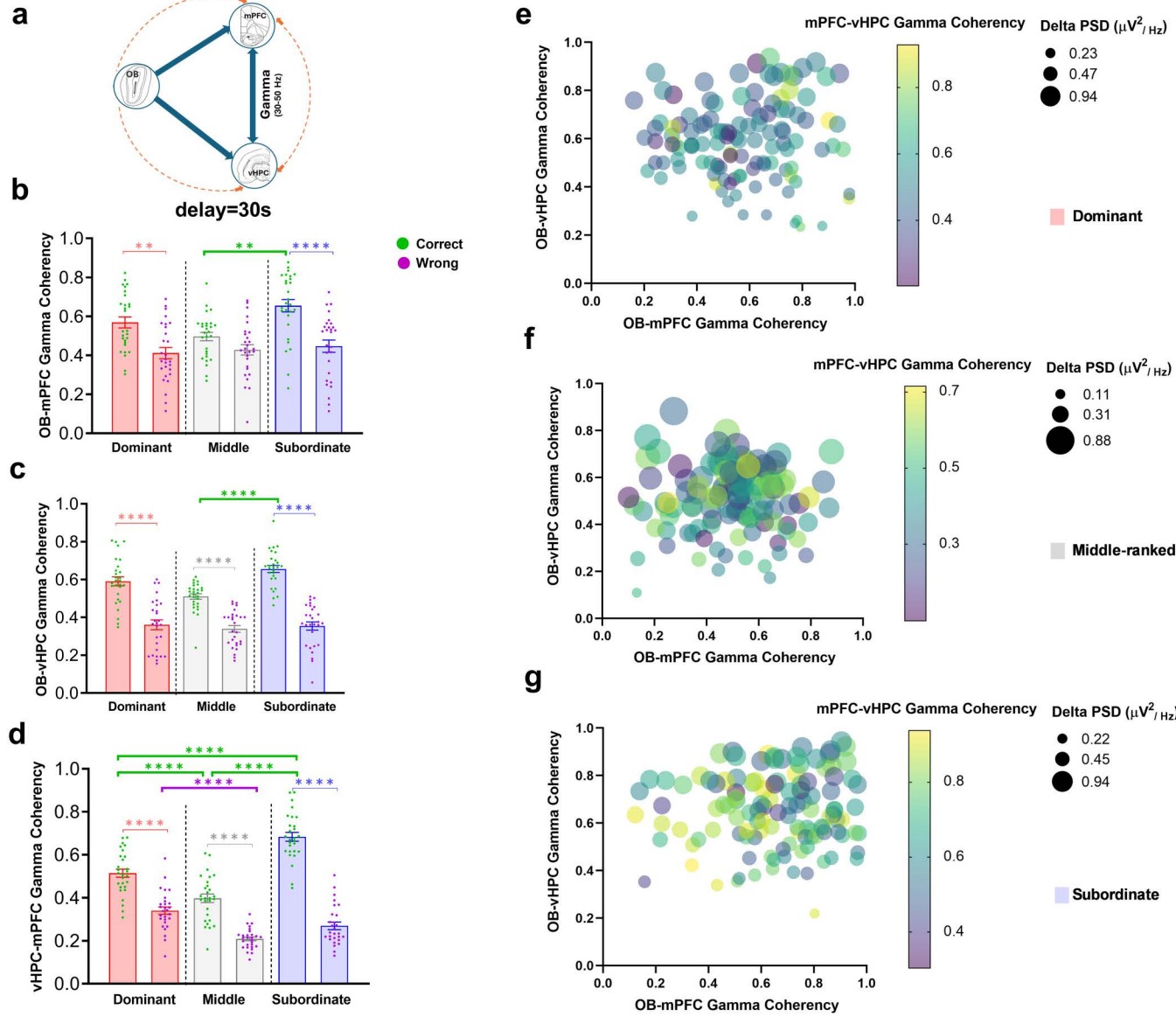

**Fig 5. Gamma coherence distribution during correct and wrong responses in the easy level of SWM task (delay = 30s).** (a) schematic illustration of OB-vHPC-mPFC circuit connections. Comparative diagram of gamma coherence in (b) OB-mPFC, (c) OB-vHPC, and (d) vHPC-mPFC circuits during correct and wrong responses; Values are expressed as mean ±SEM. Scatter plots represent the distribution of gamma range coherence in the OB-mPFC, OB-vHPC, and vHPC-mPFC circuits relative to mean delta PSD of the OB for (e) dominant, (f) middle-ranked, and (g) subordinate groups (OB: olfactory bulb; mPFC: medial prefrontal cortex; vHPC: ventral hippocampus; PSD: power spectral density). *$p < 0.05$, **$p < 0.01$, ***$p < 0.001$, ****$p < 0.0001$.

The scatter plots (Fig 5 f–h) represent the various patterns in the distribution of gamma range coherencies in the OB-mPFC, OB-vHPC, and vHPC-mPFC in respect to mean delta PSD of the OB during *easy level of task* for dominant, middle-ranked, and subordinate groups, respectively. Data points are closely clustered across all three social ranks, indicating minimal differences in neural synchronization under low cognitive load. No clear separation is observed between dominant, middle-ranked, and subordinate groups, suggesting that these neural circuits operate similarly across ranks in

the easy level of task. These plots serve as a visual complement to the statistical analyses, showing that gamma synchrony differences between social groups are minimal when task difficulty is low.

**3.2.5. Gamma coherence in OB-mPFC, OB-vHPC, and vHPC-mPFC circuits during correct and wrong responses in the difficult level of SWM task across different social ranks.** In *the difficult level of task*, gamma coherence in the OB-mPFC, OB-vHPC, and vHPC-mPFC circuits during correct and wrong responses across social ranks were analyzed using two-way ANOVA.

**3.2.5.1. OB-mPFC coherence.** A significant main effect of response type was found ($F_{(1, 162)} = 27.81$, $P < 0.0001$), while neither group ($F_{(2, 162)} = 0.65$, $P = 0.52$) nor the interaction ($F_{(2, 162)} = 1.85$, $P = 0.16$) effects reached significance (Fig 6b). Post-hoc Bonferroni comparisons showed that gamma coherence during correct responses was significantly higher than wrong responses in middle-ranked ($t_{(162)} = 3.61$, $P < 0.01$) and subordinate ($t_{(162)} = 4.03$, $P < 0.01$) animals. All three social ranks exhibited similar gamma coherence during correct trials (Fig 6b).

**3.2.5.2. OB-vHPC coherence.** Significant main effects of group ($F_{(2, 162)} = 7.13$, $P < 0.01$) and response type ($F_{(1, 162)} = 69.32$, $P < 0.0001$) were observed, with no significant interaction ($F_{(2, 162)} = 0.73$, $P = 0.47$; Fig 6c). Bonferroni post-hoc tests revealed significantly higher coherence during correct versus wrong responses across all social ranks (dominant: $t_{(162)} = 5.73$, $P < 0.0001$; middle-ranked: $t_{(162)} = 4.63$, $P < 0.001$; subordinate: $t_{(162)} = 4.04$, $P < 0.01$). Gamma coherence during correct trials was similar among social ranks (Fig 6c).

**3.2.5.3. vHPC-mPFC coherence.** Main effects of group ($F_{(2, 162)} = 5.94$, $P < 0.01$) and response type ($F_{(1, 162)} = 16.33$, $P < 0.0001$) were significant; interaction was not ($F_{(2, 162)} = 2.27$, $P = 0.10$; Fig 6d). Bonferroni comparisons showed higher gamma coherence during correct compared to wrong responses in dominant ($t_{(162)} = 3.38$, $P < 0.05$) and subordinate ($t_{(162)} = 3.0$, $P < 0.05$) animals. Additionally, during correct trials, subordinates showed significantly higher coherence than middle-ranked animals ($t_{(162)} = 3.11$, $P < 0.05$; Fig 6d).

The scatter plots (Fig 6 f–h) represent the various patterns in the distribution of gamma range coherenc in the OB-mPFC, OB-vHPC, and vHPC-mPFC in respect to mean delta PSD of the OB during difficult level of task for dominant, middle-ranked, and subordinate groups, respectively. In *the difficult level of task*, the gamma synchrony patterns clearly different. The dominant and middle-ranked showed lower levels of synchrony, while the Subordinate group experienced the highest levels in both neural pathways. This observation suggests that the pattern of neural communication in this brain circuit changes depending on social status and cognitive load. The details of these differences are discussed in the Discussion section.

**3.2.6. Comparison of gamma coherence in OB-mPFC, OB-vHPC, and vHPC-mPFC circuits during correct responses in the easy and difficult levels of SWM task across different social ranks.** Comparison of gamma coherence during correct responses between *two levels of task difficulty* across social ranks was performed using two-way ANOVA.

**3.2.6.1. OB-mPFC coherence.** Significant main effects of group ($F_{(2, 162)} = 7.95$, $P < 0.001$), task difficulty ($F_{(1, 162)} = 16.69$, $P < 0.0001$), and their interaction ($F_{(2, 162)} = 3.22$, $P < 0.05$) were observed (Fig 7b). Bonferroni post-hoc tests showed significantly higher coherence in subordinates during the easy level of task compared to the difficult level ($t_{(162)} = 3.78$, $P < 0.01$; Fig 7b).

**3.2.6.2. OB-vHPC coherence.** There were significant main effects of group ($F_{(2, 162)} = 11.10$, $P < 0.0001$) and task difficulty ($F_{(1, 162)} = 8.38$, $P < 0.01$), but no significant interaction effect ($F_{(2, 162)} = 0.98$, $P = 0.37$; Fig 7c). Bonferroni comparisons revealed no significant differences in gamma coherence between easy and difficult levels of task.

**3.2.6.3. vHPC-mPFC coherence.** Main effects of group ($F_{(2, 162)} = 40.00$, $P < 0.0001$), task difficulty ($F_{(1, 162)} = 72.89$, $P < 0.0001$), and interaction ($F_{(2, 162)} = 10.83$, $P < 0.0001$) were significant (Fig 7d). Bonferroni post-hoc tests indicated significantly higher gamma coherence in dominants ($t_{(162)} = 4.67$, $P < 0.0001$) and subordinates ($t_{(162)} = 8.33$, $P < 0.0001$; Fig 7d) during the easy level compared to the difficult level of task.

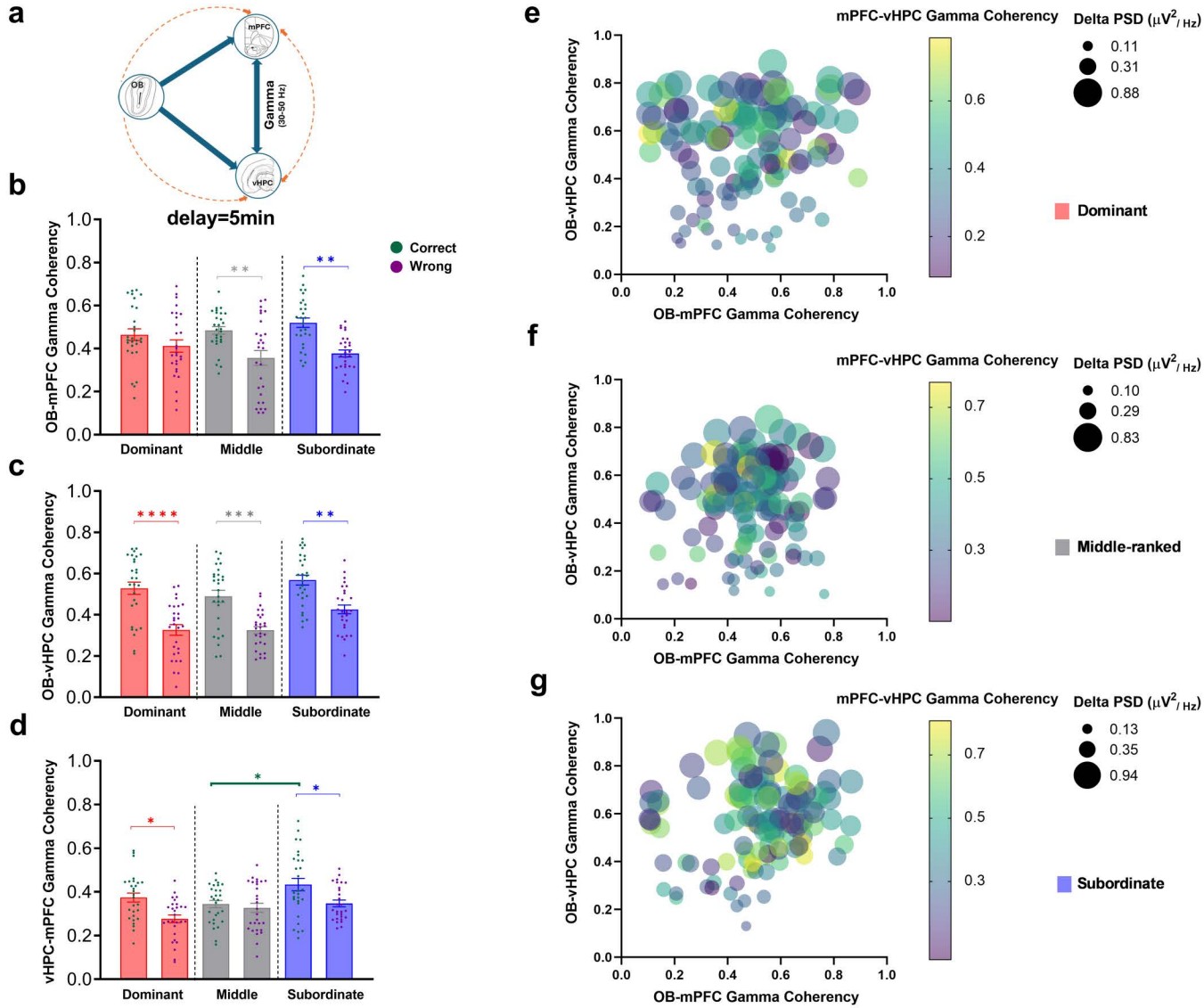

**Fig 6. Gamma coherence distribution during correct and wrong responses in the difficult level of SWM task (delay: 5 min).** (a) schematic illustration of OB-vHPC-mPFC circuit connections. Comparative diagram of gamma coherence of (b) OB-mPFC, (c) OB-vHPC, and (d) vHPC-mPFC circuits during correct and wrong responses; Values are expressed as mean ±SEM. Scatter plots show the distribution of gamma range coherence in the OB-mPFC, OB-vHPC, and vHPC-mPFC circuits relative to mean delta PSD of the OB for (e) dominant, (f) middle-ranked, and (g) subordinate groups (OB: olfactory bulb; mPFC: medial prefrontal cortex; vHPC: ventral hippocampus; PSD: power spectral density). *p < 0.05, **p < 0.01, ***p < 0.001, ****p < 0.0001.

**3.2.7. Correlation between OB delta PSD and vHPC-mPFC gamma coherence during correct responses in the easy and difficult levels of SWM task across different social ranks.** A multiple regression analysis was conducted to assess the influence of OB delta PSD and groups on gamma coherence of the vHPC-mPFC circuit during *two levels of task difficulty*. In *the easy level of task*, the overall regression model was significant (F (5, 354) = 48.98, P < 0.0001). The effect of OB delta PSD (F (1, 354) = 6.27, P = 0.01) and the interaction between groups and OB delta PSD (F (2, 354) = 15.88, P < 0.0001) were significant. While the main effect of group was not significant (F (2, 354) = 0.88, P = 0.41; Fig

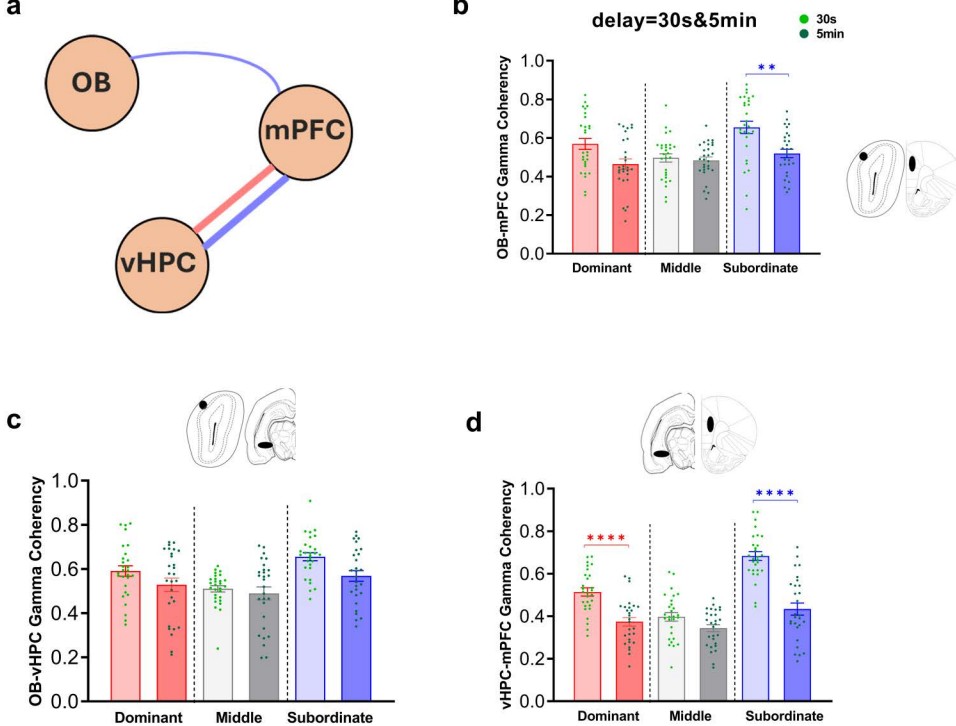

**Fig 7. Gamma coherence distribution during correct responses of easy and difficult levels of SWM task (delays: 30s & 5 min).** (a) schematic illustration of OB-vHPC-mPFC circuit connections. Comparative diagram of gamma coherence in (b) OB-mPFC, (c) OB-vHPC, and (d) vHPC-mPFC circuits during correct responses of easy and difficult levels of task; Values are expressed as mean ±SEM. (OB: olfactory bulb; mPFC: medial prefrontal cortex; vHPC: ventral hippocampus; PSD: power spectral density). *$**p < 0.01$, $****p < 0.0001$.*

8a). These results show that the OB delta PSD and groups significantly predicted gamma coherence in the vHPC-mPFC circuit. Correlation analysis showed that increased OB delta rhythm was positively correlated with vHPC-mPFC gamma coherence during correct responses in the dominant ($r = 0.22$, $P = 0.01$) and subordinate ($r = 0.43$, $P < 0.0001$) groups, but negatively correlated in the middle-ranked animals ($r = -0.22$, $P = 0.01$; Fig 8a).

In *the difficult level of task*, the overall regression model was again significant ($F_{(5, 354)} = 6.10$, $P < 0.0001$), but neither the main effects of groups ($F_{(2, 354)} = 0.63$, $P = 0.53$), OB delta PSD ($F_{(1, 354)} = 2.63$, $P = 0.10$), nor their interaction ($F_{(2, 354)} = 1.02$, $P = 0.36$; Fig 8b) were significant. This suggests that OB delta PSD did not predict gamma coherence of the vHPC-mPFC circuit during the difficult level of task. Correlation analysis revealed a negative association between OB delta PSD and vHPC-mPFC gamma coherence during correct responses in all social ranks, which was significant only for middle-ranked animals (dominant: $r = -0.15$, $P = 0.08$; middle-ranked: $r = -0.28$, $P = 0.001$; subordinate: $r = -0.11$, $P = 0.20$; Fig 8b).

Furthermore, cross-correlation analysis was performed to further investigate the temporal relationship between OB delta oscillations and gamma oscillations in the mPFC and vHPC. The full results of this analysis, including statistical data and corresponding figures, are presented in the Supporting Information (S1 File).

## 4. Discussion

We investigated whether delta oscillations in the OB synchronize the gamma rhythm in vHPC-mPFC circuit activity during two levels of difficulty using a SWM task in male rats with different social ranks (i.e., dominant, middle-ranked,

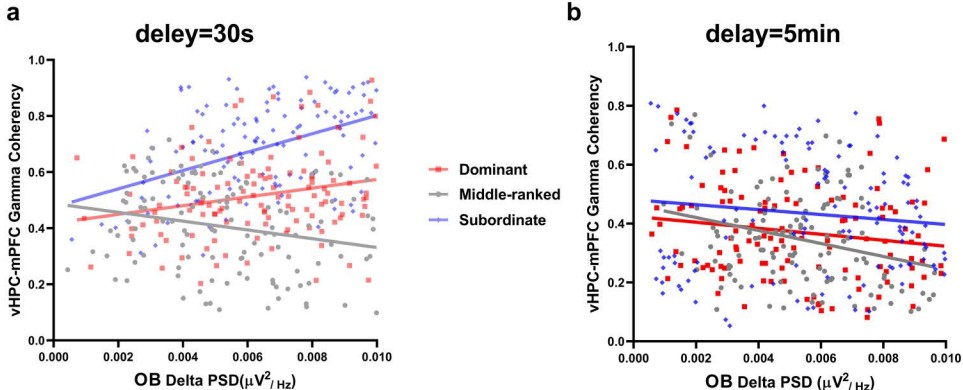

**Fig 8. Correlation analysis between OB delta PSD and vHPC-mPFC gamma coherence during correct responses. (a)** The easy level of task. **(b)** The difficult level of task (OB: olfactory bulb; mPFC: medial prefrontal cortex; vHPC: ventral hippocampus; PSD: power spectral density).

and subordinate). Our study demonstrates that subordinate rats exhibit enhanced SWM performance in the easy level of task, as evidenced by shorter latencies, alongside heightened delta oscillations in the OB and increased gamma coherence in the vHPC-mPFC circuit. These results underscore the dynamic interplay between SDH, olfactory processing, and memory-related neural circuits.

### 4.1. Behavioral implications of social hierarchy

SDH has been linked to differences in decision speed, with dominant and subordinate animals often exhibiting different response latencies depending on task demands [3]. Interestingly, subordinate rats frequently demonstrate superior memory performance, which stands in contrast to the prevailing notion that subordinates generally possess weaker cognitive abilities. These observations underscore the importance of accounting for multiple potential confounding factors—such as genetic background, unequal access to resources, mating experiences, and post-hierarchy housing conditions—most of which were carefully controlled in the present study [45–47]. Contrary to conventional assumptions that subordinate animals exhibit poorer cognitive performance, in line with previously reported behavioral data, our analyses show that subordinates outperformed dominants in the easy level of SWM task. Subordinate rats demonstrated shorter latencies in the easy level of SWM task compared to dominant and middle-ranked animals, despite no significant differences in the percentage of correct responses [25]. This aligns with recent findings on SWM performance following hierarchy formation, suggesting adaptive cognitive strategies in subordinates to compensate for resource limitations [23]. Such adaptations may involve prioritizing rapid decision-making in low-demand task, as seen in their shorter latencies. This phenomenon parallels findings in humans, where lower social status correlates with enhanced vigilance and faster responses in specific contexts [6]. The lack of significant differences in correct responses across ranks, however, highlights that social hierarchy may influence processing speed rather than accuracy under low cognitive load.

### 4.2. OB delta oscillations as a synchronizer of SWM circuits

The elevated OB delta PSD (<4 Hz) during correct responses in dominant and subordinate rats highlights the critical role of olfactory-respiratory rhythms in SWM. Delta oscillations in the OB are tightly coupled to nasal respiration [28,48], which entrains rhythmic activity in downstream limbic and cortical regions [48,49]. Our results extend prior work by demonstrating that OB delta activity correlates with vHPC-mPFC gamma coherence in subordinate rats, suggesting that respiratory-entrained rhythms facilitate cross-regional synchrony for optimal SWM performance [15]. Specifically, the AON, which projects directly to the mPFC [50], may relay olfactory signals to enhance

gamma-band coordination during spatial decision-making [30]. Subordinates, which rely heavily on olfactory cues for environmental navigation and social communication [51], may prioritize this pathway to compensate for their lower social rank. Multiple linear regression analysis showed that both a higher percentage of correct responses and a shorter delay during the SWM task significantly predicted increased delta PSD in the OB, and these effects were rank-dependent. In subordinates, delta PSD was positively correlated with correct responses and negatively correlated with latency in both easy and difficult levels of task, suggesting that stronger delta rhythms are associated with faster and more accurate decisions. In contrast, dominant rats showed a consistent negative relationship between OB delta PSD and correct responses, particularly in the difficult level of task indicating possible task on non-olfactory strategies. Middle-ranked rats showed the opposite pattern in the easy level of task, where delta PSD was positively correlated with latency and negatively correlated with correct responses, which may indicate less effective or more deliberate decision-making strategies. Thus, task difficulty modulated these relationships. In the easy level of task, differences in delta PSD between correct and wrong responses were significant in dominants and subordinates but not in middle-ranked rats, while in the difficult level of task, group differences largely disappeared, although the correlation patterns persisted. The maintenance of a positive relationship between delta oscillations and correct responses in subordinates, and a negative relationship in dominants at both levels of task difficulty, indicates an interaction in OB-dependent pathways that depends on social rank. Moreover, delta PSD in the OB significantly predicted gamma-band coherence in the vHPC-mPFC circuit during successful SWM task performance, with these associations varying across SDH ranks. The negative correlation between OB delta PSD and vHPC-mPFC coherence in middle-ranked rats during the easy level of task suggests divergent neural strategies across social ranks. Middle-ranked animals, often in unstable hierarchical positions [30], may engage alternative circuits, such as the dorsal hippocampus or amygdala, to navigate spatial challenges, a hypothesis requiring further exploration. Interestingly, during the difficult levels of task, subordinate rats exhibited increased OB–mPFC delta–gamma cross-correlation in wrong trials, despite higher delta PSD and coherence during correct responses (S1 File). This may indicate excessive or non-optimal synchrony in the OB–mPFC circuit associated with unsuccessful decision-making.

### 4.3. Neural synchrony and social rank: Gamma coherence dynamics in memory circuits

Gamma coherence within the OB-mPFC, OB-vHPC, and vHPC-mPFC circuits revealed distinct patterns depending on task difficulty and social rank. During the easy working memory task, gamma coherence was significantly higher during correct compared to incorrect responses across all social ranks, with the dominant and subordinate groups showing particularly pronounced differences. These results highlight the essential contribution of gamma synchrony in facilitating efficient SWM processing by enhancing synaptic coordination and temporal precision within key memory-related circuits [52,53]. Subordinate rats exhibited greater gamma coherence than middle-ranked and dominant rats during correct trials, suggesting a compensatory enhancement of neural synchrony in lower-ranked animals, potentially reflecting an increased reliance on olfactory processing and network integration to offset social status disadvantages [54,55]. This enhanced gamma coordination in subordinates may facilitate rapid and accurate decision-making under low cognitive demand. In difficult level of task, although gamma coherence remained higher during correct responses across all ranks, the differences between social ranks diminished or disappeared. This attenuation may be attributed to the increased cognitive load and complexity, which can disrupt hippocampal-prefrontal synchrony and reduce the influence of social hierarchy on neural circuit dynamics [20,56]. The uniform reduction in gamma coherence across groups suggests recruitment of alternative compensatory mechanisms, such as increased theta-gamma coupling or engagement of other sensory modalities, to meet the heightened task demands [57,58]. Scatter plot analyses further highlight rank-dependent patterns of neural synchrony. Subordinates maintained the highest gamma coherence levels in both levels of task difficulty, particularly in the OB-vHPC and vHPC-mPFC pathways, reinforcing the idea of adaptive neural plasticity and circuit reorganization in response to social rank and cognitive load.

### 4.4. Task difficulty synchronizes hierarchy-dependent neural dynamics

Comparison of the *two difficulty levels* revealed that increased task demand was accompanied by reduced gamma rhythm coordination within the OB–vHPC–mPFC network. This may be due to decreased motivation to select the correct arm under challenging conditions, leading to more random arm choices and diminishing the differences in performance across SDH. [59]. The lack of rank-dependent differences in gamma coherence during the difficult level of task corresponds with the behavioral data indicating no significant differences in correct responses across different SDH, supporting the idea of diminished task engagement or altered strategy under higher cognitive load (5-minute delay). The disappearance of rank-specific differences in the difficult level of task (5-minute delay) aligns with studies showing that prolonged delays disrupt hippocampal-prefrontal synchrony [16,60]. Under high cognitive load, the mPFC may shift from integrating spatial information to managing interference or updating task rules [61], reducing the advantage conferred by OB-driven delta-gamma coupling. The absence of significant interaction effects in OB-vHPC coherence implies that this circuit may be less sensitive to the combined influence of social hierarchy and task difficulty compared to other pathways. The uniform reduction in vHPC-mPFC gamma coherence across all ranks during the difficult level of task further supports this, indicating that compensatory mechanisms (e.g., increased theta-gamma coupling or reliance on alternate sensory modalities) are recruited to meet heightened cognitive demands [25,26].

### 4.5. Anxiety, structural plasticity, and social rank

Structural and emotional differences across ranks may further explain our findings. Subordinate rats exhibit greater dendritic spine density in the mPFC and hippocampal CA1 [62], which could enhance synaptic plasticity and circuit efficiency during SWM. Conversely, dominant rats display higher anxiety-like behaviors [63], which may prolong risk assessment and delay decision-making. However, the tube test's minimal impact on anxiety [55] and previous reports showing no significant differences in locomotor activity across different social ranks in open field and Y-maze tests [63,64] suggest that our behavioral outcomes primarily reflect cognitive and structural adaptations rather than baseline activity or general emotional differences.

## 5. Limitations

Certain limitations need to be acknowledged when interpreting the findings of the present study. The cross-frequency coupling analysis was not performed; therefore, potential causal interactions across frequency bands were not examined. Moreover, the SWM task in this study was performed without rewards, a protocol deliberately chosen to minimize potential confounding effects of differential food motivation between dominant and subordinate rats [38]. Nevertheless, the current findings are limited to the T-maze paradigm, and conducting the same study without rewards using other working memory tasks could provide broader validation of the results and clarify the interaction between working memory circuits within the framework of social hierarchy.

## 6. Future directions

We suggest further future studies of pharmacological intervention of OB region to clarify causal circuit-level mechanisms during SWM task in rats with different social ranks. Such interventions could reveal whether modulating OB activity can alter its role in the vHPC-mPFC circuit during working memory processing in rats with various dominance ranks. On the other hand, future studies using cross-frequency coupling analysis and some other SWM tests may clarify the relationship between the neural circuits involved and the cognitive effects of SDH. Future research on SDH and cognitive function is strongly encouraged to employ extended cognitive assessments to validate these findings.

## 7. Conclusion

Overall, the present study demonstrates that subordinate rats exhibited shorter latencies to reach the goal arm compared to dominant rats in a short-delay SWM task using T-maze spontaneous alternations. Moreover, we demonstrated that OB delta oscillatory activity can synchronize the activity of the vHPC-mPFC circuit and increase gamma coherence during

easy level of SWM task in subordinate rats to improve their performance. As cognitive load increases, this advantage diminishes, reflecting the context-dependent nature of hierarchy-related neural adaptations. These findings deepen our understanding of how social status shapes sensory-cognitive integration and highlight the OB as a pivotal node in the synchronization of memory circuits.

## Supporting information

**S1 File. Cross-correlation between OB delta and gamma oscillations in mPFC (a–c) and vHPC (d–f) during correct and wrong responses across two levels of task difficulty and different social ranks.** (a) Easy level of task (OB–mPFC), (b) Difficult level of task (OB–mPFC; ***$p < 0.001$, Values are expressed as mean ±SEM.), (c) Easy vs. difficult levels of task (OB–mPFC), (d) Easy level of task (OB–vHPC), (e) Difficult level of task (OB–vHPC), (f) Easy vs. difficult levels of task (OB–vHPC).
(ZIP)

## Author contributions

**Conceptualization:** Soomaayeh Heysieattalab, Ali Jaafari suha.

**Formal analysis:** Elham Bakhshi Jifroudi, Farhad Farkhondeh Tale Navi.

**Investigation:** Elham Bakhshi Jifroudi, Faezeh Zarfsaz.

**Methodology:** Soomaayeh Heysieattalab, Faezeh Zarfsaz.

**Resources:** Yousef Panahi.

**Supervision:** Soomaayeh Heysieattalab.

**Validation:** Soomaayeh Heysieattalab.

**Visualization:** Elham Bakhshi Jifroudi, Farhad Farkhondeh Tale Navi.

**Writing – original draft:** Elham Bakhshi Jifroudi.

**Writing – review & editing:** Soomaayeh Heysieattalab, Farhad Farkhondeh Tale Navi, Ali Jaafari suha.

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
