## [Decision Letter · Decision Letter 0]

23 Nov 2025

Dear Dr. Heysieattalab,

Thank you for submitting your manuscript to PLOS ONE. After careful consideration, we feel that it has merit but does not fully meet PLOS ONE’s publication criteria as it currently stands. Therefore, we invite you to submit a revised version of the manuscript that addresses the points raised during the review process.

**ACADEMIC EDITOR:** please do follow advices from our reviewers

We look forward to receiving your revised manuscript.

Kind regards,

Prof. Dr. Dragan Hrncic, MD, PhD

Academic Editor

PLOS ONE

Journal Requirements:

2. To comply with PLOS One submissions requirements, in your Methods section, please provide additional information regarding the experiments involving animals and ensure you have included details on (1) methods of sacrifice, (2) methods of anesthesia and/or analgesia, and (3) efforts to alleviate suffering.

4. In the online submission form, you indicated that [Data will be made available upon reasonable request from the corresponding author.].

5. Please upload a new copy of Figure 1 as the detail is not clear. Please follow the link for more information:  https://journals.plos.org/plosone/s/figures

6. We notice that your supplementary figure (Figure S1) is uploaded with the file type 'Figure'. Please amend the file type to 'Supporting Information'. Please ensure that each Supporting Information file has a legend listed in the manuscript after the references list.

Reviewers' comments:

Reviewer's Responses to Questions

**Comments to the Author**

1. Is the manuscript technically sound, and do the data support the conclusions?

Reviewer #1: Yes

2. Has the statistical analysis been performed appropriately and rigorously?

Reviewer #1: Yes

3. Have the authors made all data underlying the findings in their manuscript fully available?

Reviewer #1: Yes

4. Is the manuscript presented in an intelligible fashion and written in standard English?

Reviewer #1: Yes

Reviewer #1: Very interesting and well-written article that reflects a well-designed and carefully conducted study. It is suitable for publication.

The introduction is rather extensive for a scientific article, containing a detailed description of the current knowledge in the field. A significant reduction of the Introduction section is recommended

Attention should be given to the quality of the figures. The images in the PDF version of the manuscript appear to have low resolution.

Throughout the text, the formatting of statistical values should be standardized. For example, ensure consistency between 'P=0.005' and 'P = 0.005’

Best regards,

**Do you want your identity to be public for this peer review?** For information about this choice, including consent withdrawal, please see our Privacy Policy

Reviewer #1: **Yes:**  Jorge Miranda Rodrigues

---

## [Author Response · Author response to Decision Letter 1]

2 Dec 2025

Manuscript Number.: PONE-D-25-54057

Olfactory Bulb Differently Synchronizes Ventral Hippocampus–Medial Prefrontal Cortex Circuit During Spatial Working Memory Across Social Dominance Hierarchies

Editor and Reviewer comments:

We sincerely thank the editor and the reviewer for the time and thoughtful consideration they dedicated to evaluating our manuscript. We have carefully addressed all comments and revised the manuscript accordingly. We believe that the revisions made in response to their constructive feedback have significantly improved the quality and clarity of the work. Below, we provide a point-by-point response to each comment.

#Journal Requirements:

1. Please ensure that your manuscript meets PLOS ONE's style requirements, including those for file naming. The PLOS ONE style templates can be found at 1. Please ensure that your manuscript meets PLOS ONE's style requirements, including those for file naming. The PLOS ONE style templates can be found at

Thank you for these clear guidelines. We have revised the manuscript to fully comply with PLOS ONE's style requirements. All changes are clearly indicated throughout manuscript.

2. To comply with PLOS One submissions requirements, in your Methods section, please provide additional information regarding the experiments involving animals and ensure you have included details on (1) methods of sacrifice, (2) methods of anesthesia and/or analgesia, and (3) efforts to alleviate suffering.

The Methods section has been updated to include further information on animal procedures, including the method of sacrifice, anesthesia and analgesia protocols, and measures taken to minimize animal distress. These additions are included in the “Electrode Implantation and Histological Verification” subsection.

#2.2.2.1. Electrode Implantation and Histological Verification: Anesthesia depth was continuously monitored by assessing tail and paw pinch reflexes. Additionally, vitamin A ointment was applied to the eyes to prevent corneal drying during surgery. Local anesthesia of the scalp was provided by subcutaneous injection of 0.5 ml Persocaine to minimize pain during the incision.

After the completion of experiments, to ensure accurate electrode placement, rats were deeply anesthetized with carbon dioxide. Once breathing ceased completely and the animals exhibited ocular pallor, they were rapidly decapitated using a guillotine. Brains were carefully extracted and fixed in 4% paraformaldehyde at 4 °C for 48 h. Subsequently, the brains were sectioned using a vibroslicer, stained with methylene blue, and examined under a microscope (AC 230V 50 Hz, Fig. 1b).

Thank you for your comment. I would like to inform you that I have now made all the data publicly available in accordance with the journal’s open data policy.

#Data availability: The data associated with this research are openly accessible on the Open Science Framework (OSF) platform via the following link:

https://osf.io/sjex3/?view_only=c0da26cdab6e4141995620a703a016d8

4. In the online submission form, you indicated that [Data will be made available upon reasonable request from the corresponding author.].

In accordance with PLOS ONE policy, all relevant data have been deposited in the Open Science Framework (OSF) repository and are now publicly accessible.

5. Please upload a new copy of Figure 1 as the detail is not clear. Please follow the link for more information: https://journals.plos.org/plosone/s/figures

Thank you. A new, high-resolution version of Fig 1 has been uploaded according to the provided figure preparation guidelines.

6. We notice that your supplementary figure (Figure S1) is uploaded with the file type 'Figure'. Please amend the file type to 'Supporting Information'. Please ensure that each Supporting Information file has a legend listed in the manuscript after the references list.

Thank you for your helpful comment. We have amended the file type of Figure S1 to “S1_File” and uploaded it as a single ZIP file named “Supporting Information.” In addition, the corresponding legend in the manuscript has been updated as follows:

File. Cross-correlation between OB delta and gamma oscillations in mPFC (a–c) and vHPC (d–f) during correct and wrong responses across two levels of task difficulty and different social ranks. (a) Easy level of task (OB–mPFC), (b) Difficult level of task (OB–mPFC; ***p < 0.001, Values are expressed as mean ±SEM.), (c) Easy vs. difficult levels of task (OB–mPFC), (d) Easy level of task (OB–vHPC), (e) Difficult level of task (OB–vHPC), (f) Easy vs. difficult levels of task (OB–vHPC).

Thank you for your guidance. We have prepared a 'Revised Manuscript with Track Changes' in which all modifications related to the Supporting Information captions and corresponding in-text citations are clearly highlighted.

Thank you for your helpful recommendation. We have carefully reviewed the reference list and confirm that none of the cited articles have been retracted.

#Reviewer 1:

1. The introduction is rather extensive for a scientific article, containing a detailed description of the current knowledge in the field. A significant reduction of the Introduction section is recommended

Thank you for your valuable suggestion. We have carefully revised and shortened the Introduction section. All modifications related to this comment are clearly highlighted in the 'Revised Manuscript with Track Changes' for your reference.

2. Attention should be given to the quality of the figures. The images in the PDF version of the manuscript appear to have low resolution.

Thank you for your comment. We have replaced all figures with higher-resolution versions to ensure clear visibility in the PDF version of the manuscript.

3. Throughout the text, the formatting of statistical values should be standardized. For example, ensure consistency between 'P=0.005' and 'P = 0.005’

Thank you for your helpful comment. We have standardized the formatting of all statistical values throughout the manuscript. All corresponding edits can be seen in the 'Revised Manuscript with Track Changes'.

---

## [Editor Report · Decision Letter 1]

4 Jan 2026

Olfactory Bulb Differently Synchronizes Ventral Hippocampus–Medial Prefrontal Cortex Circuit During Spatial Working Memory across Social Dominance Hierarchies

PONE-D-25-54057R1

Dear Dr. Heysieattalab,

We’re pleased to inform you that your manuscript has been judged scientifically suitable for publication and will be formally accepted for publication once it meets all outstanding technical requirements.

Kind regards,

Prof. Dr. Dragan Hrncic, MD, PhD

Academic Editor

PLOS One
---

## [Editor Report · Acceptance letter]

PONE-D-25-54057R1

PLOS One

Dear Dr. Heysieattalab,

I'm pleased to inform you that your manuscript has been deemed suitable for publication in PLOS One. Congratulations! Your manuscript is now being handed over to our production team.

Kind regards,

on behalf of

Professor Dragan Hrncic

Academic Editor

PLOS One